# Regularizing by the Variance of the Activations' Sample-Variances

**Etai Littwin**[1]    **Lior Wolf** [1,2]
[1]Tel Aviv University    [2]Facebook AI Research

## Abstract

Normalization techniques play an important role in supporting efficient and often more effective training of deep neural networks. While conventional methods explicitly normalize the activations, we suggest to add a loss term instead. This new loss term encourages the variance of the activations to be stable and not vary from one random mini-batch to the next. As we prove, this encourages the activations to be distributed around a few distinct modes. We also show that if the inputs are from a mixture of two Gaussians, the new loss would either join the two together, or separate between them optimally in the LDA sense, depending on the prior probabilities. Finally, we are able to link the new regularization term to the batchnorm method, which provides it with a regularization perspective. Our experiments demonstrate an improvement in accuracy over the batchnorm technique for both CNNs and fully connected networks.

## 1   Introduction

We propose a novel regularization technique that is applied before the activation of all neurons in the neural network. The new regularization term encourages the distribution of the individual activations to have a few distinct modes. This property is achieved implicitly by computing the variance of the activation of each neuron in each minibatch and by penalizing for variations of this variance, i.e., we encourage the variances to be the same across the mini-batches.

We provide a theoretical link between the variance-based regularization term and the resulting peaked activation distributions, which we also observe experimentally, see Fig. 1. In addition, we also provide experimental evidence that the new term leads to improved accuracy and can replace, during training, normalization techniques such as the batch-norm technique.

The link between the new regularization term and batch-norm is further explored theoretically. A distribution with few modes would lead to more stable batches and, for example, the representation of a given sample would not vary along different batches. In other words, it is desirable that a sample, if repeated twice in multiple batches, would produce the same network activations post-normalization. This is an indirect way in which batchnorm benefits from few-modes. In our method it is encouraged more explicitly.

The new regularization term is adaptive, in the sense that it can lead to a few distinct outcomes. When applied to a mixture of two Gaussians, the regularization leads, in an unsupervised way, to one of two possible projections: either the LDA projection that maximally separates between the two Gaussians, or the orthogonal projection that is least sensitive to their differences.

Interestingly, the amount of variance in each activation is controlled by a parameter $\beta$. In order to avoid searching over a wide range of hyper-parameters, we optimize for this term during training and allow each neuron to adapt to a different level of variance.

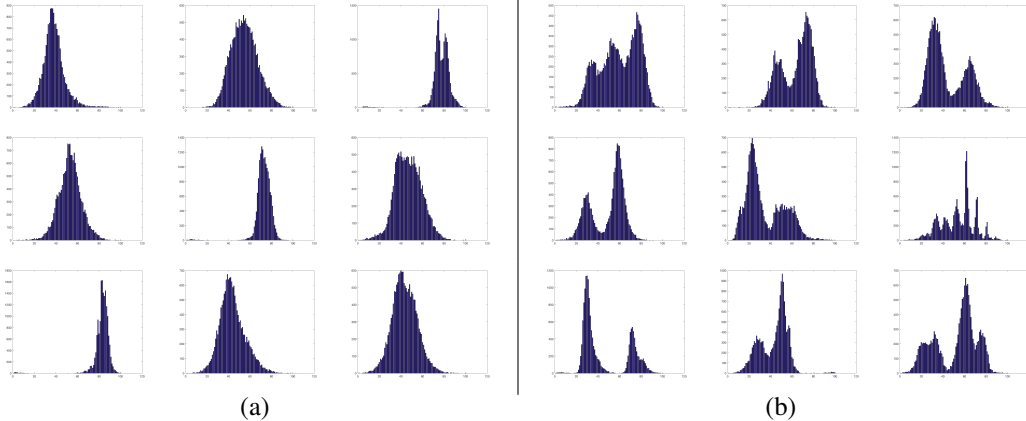

(a)            (b)

Figure 1: Histograms of activations in a network trained on the UCI adult dataset. (a) Random neurons trained with batchnorm. (b) Random neurons trained with our VCL method. Each row corresponds to a different hidden layer.

## 2 The Variance Constancy Loss

The distribution of the activations of each neuron depends on both the distribution of network inputs and the weight of the network upstream from that neuron. Let $\rho$ be a random variable denoting the activations of a single neuron and denote the underlying distribution as $p$. The variance of $\rho$ is given by $\sigma^2 = \mathbb{E}[(\rho - \mu_\rho)^2]$, where $\mu_\rho = \mathbb{E}[\rho]$. For a finite sample $s = \{\rho_1...\rho_n\}$ randomly drawn from $p$, the unbiased sample variance of $p$ over $s$ is given by $\sigma_s^2 = \frac{1}{n-1} \sum_{i=1}^{n} (\rho_{s_i} - \frac{1}{n} \sum_{i=1}^{n} \rho_{s_i})^2$. The variance of the sample variances is given by:

$$\mathbb{E}[(\sigma^2 - \sigma_s^2)^2] = \frac{m_4}{n} - \frac{\sigma^4(n-3)}{n(n-1)} \tag{1}$$

where $m_4 = \mathbb{E}[(\rho - \mu_\rho)^4]$ is the fourth moment of $\rho$ [4].

From Eq. 1, given that the distribution has a given variance, the variance of variance is controlled by $n$ and the fourth moment of the distribution. We would like to show that this variance of measured variances is low for distributions with few modes. Intuitively, a distribution with a few distinct modes would have a low variance of sample variance, since there is a relatively small number of possibilities to sample from. Consider, for example, a distribution of 2 modes and a sample size of $n$. There are only $n$ possible patterns to select from the two modes. For $n = 3$ there are $aaa$, $aab$, $abb$, and $bbb$, where $a$ and $b$ represent selecting from the first mode or from the second mode. For a distribution with $k$ modes, this would be $\binom{n+k-1}{k}$, which can be considerably larger.

In the following analysis we characterize distributions with low variance of sample variance. Specifically, we are interested in distributions $p_\rho$ such that the quantity $\mathbb{E}[(\sigma^2 - \sigma_s^2)^2]$ is minimized under the constraint that the variance is fixed, i.e., $\sigma^2 = \alpha$. Formally, we are interested in the following minimization problem:

$$p^* = \arg\min_{p} \mathbb{E}[(\sigma^2 - \sigma_s^2)^2] \ \ s.t \ \ \sigma^2 = \alpha \tag{2}$$

Note that we can reformulate Eq. 2 as:

$$p^\star = \arg\min_{p} \mathbb{E}[(1 - \frac{\sigma_s^2}{\sigma^2})^2] \ \ s.t \ \ \sigma^2 = \alpha \tag{3}$$

The next result shows that minimizing Eq. 2 over the space of distributions will result in a distribution $p^\star$ with two modes.

**Theorem 1.** *Any minimizing distribution of Eq. 2 is of the form $\rho^\star = az + b$ such that $z$ is distributed according to the Bernoulli distribution with parameter $\frac{1}{2}$, and $a, b \in \mathbb{R}, a \neq 0$.*

*Proof.* From Eq. 3 and Eq. 1 we have:

$$p_\rho^\star = \arg\min_p \left( \frac{m_4}{\alpha^2 n} - \frac{(n-3)}{n(n-1)} \right) \tag{4}$$

and so we are left with the problem of minimizing the fourth moment of $p$ under the constraint $\sigma^2 = \alpha$.

Note that for any distribution, the variance squared is a lower bound for the fourth moment. To see this, we denote the slack variable $y = (\rho - \mu_\rho)^2$, and we have:

$$var(y) = \mathbb{E}[y^2] - (\mathbb{E}[y])^2 = m_4 - \sigma^4 \geq 0 \tag{5}$$

where equality is attained when $var(y) = 0$, i.e, when $y$ is constant. Therefore, $m_4$ is minimal when $|\rho - \mu_\rho|$ is constant, which means, since $\rho$ is not constant ($\sigma^2 = \alpha > 0$), that $p$ has two values with equal probability. $\qquad\square$

The term $\frac{m_4}{\sigma^4}$ in Eq. 4 is called kurtosis and is denoted by $\kappa(\rho)$. Distributions with high kurtosis tend to exhibit heavy tails, while distributions with low kurtosis are light tailed, with few outliers. For the two peak distribution of Thm. 1, there is no tail.

## 2.1 A Phase Shift Behavior

The condition on the variance in Eq. 3 is redundant, since neurons with fixed activations do not contribute to learning. We therefore define the variance constancy loss for a distribution $p$ as:

$$L_s(p) = \mathbb{E}[(1 - \frac{\sigma_s^2}{\sigma^2})^2] \tag{6}$$

This regularization can be seen as an additional unsupervised clustering loss per unit, since it is minimized by clustering its input to two modes. The driving force for the weights of each unit has a surprising quality, encouraging separation between clusters if they are prominent enough, or uniting the clusters if they are not, as demonstrated in the next theorem:

**Theorem 2.** *Consider the random input distributed as a GMM with two components $x \in \mathbb{R}^d \sim p\mathcal{N}(\mu_1, \Sigma_2) + (1-p)\mathcal{N}(\mu_2, \Sigma_2)$. We denote the within and between covariance matrices as $\Sigma_w = p\Sigma_1 + (1-p)\Sigma_2$, $\Sigma_b = (\mu_1 - \mu_2)(\mu_1 - \mu_2)^\top$. Given a vector of weights $\theta \in \mathbb{R}^d$, we denote $\rho = x^\top \theta$, it holds that:*

$$\arg\min_\theta \kappa(\rho) = \begin{cases} \arg\min_\theta \frac{\theta^\top \Sigma_w \theta}{\theta^\top \Sigma_b \theta} & \frac{1-\sqrt{\frac{1}{3}}}{2} \leq p \leq \frac{1+\sqrt{\frac{1}{3}}}{2} \\ \arg\min_\theta \frac{\theta^\top \Sigma_b \theta}{\theta^\top \Sigma_w \theta} & else \end{cases} \tag{7}$$

*Proof.* Note that $\rho \sim p\mathcal{N}(\mu_1^\top \theta, \theta^\top \Sigma_2 \theta) + (1-p)\mathcal{N}(\mu_2^\top \theta, \theta^\top \Sigma_2 \theta)$. For a Gaussian distribution with mean $\mu$ and variance $\sigma^2$, the non-centered fourth and second moments are given by:

$$m_4 = \mu^4 + 6\mu^2\sigma^2 + 3\sigma^4, \quad m_2 = \sigma^2 + \mu^2 \tag{8}$$

Due to the linearity of integration, the moments for a GMM distribution follows naturally. The mean of rho is given by $\mu = p\mu_1 + (1-p)\mu_2$. Noticing that $\mu_1 - \mu = (1-p)(\mu_1 - \mu_2)$, and $\mu_2 - \mu = p(\mu_2 - \mu_1)$, and denoting $p(1-p) = \alpha$, the fourth and second moments of $\rho$ are given by: $m_4 = \alpha(1 - 3\alpha)(\theta^\top \Sigma_b \theta)^2 + 6\alpha(\theta^\top \Sigma_w \theta)(\theta^\top \Sigma_b \theta) + 3(\theta^\top \Sigma_w \theta)^2$, $\sigma^2 = (\alpha\theta^\top \Sigma_b \theta + \theta^\top \Sigma_w \theta)$. and so:

$$\kappa(\rho) = \frac{\alpha(1 - 3\alpha)(\theta^\top \Sigma_b \theta)^2 + 6\alpha(\theta^\top \Sigma_w \theta)(\theta^\top \Sigma_b \theta) + 3(\theta^\top \Sigma_w \theta)^2}{((\alpha)\theta^\top \Sigma_b \theta + \theta^\top \Sigma_w \theta))^2}$$

$$= 3 + \frac{\alpha(1 - 6\alpha)(\theta^\top \Sigma_b \theta)^2}{(\alpha\theta^\top \Sigma_b \theta + \theta^\top \Sigma_w \theta)^2} \tag{9}$$

$$\arg\min_\theta \left( 3 + \frac{\alpha(1 - 6\alpha)(\theta^\top \Sigma_b \theta)^2}{(\alpha\theta^\top \Sigma_b \theta + \theta^\top \Sigma_w \theta)^2} \right) = \arg\max_\theta \frac{\alpha\theta^\top \Sigma_b \theta + \theta^\top \Sigma_w \theta}{\alpha(1 - 6\alpha)(\theta^\top \Sigma_b \theta)}$$

$$= \arg\max_\theta \frac{\theta^\top \Sigma_w \theta}{\alpha(1 - 6\alpha)(\theta^\top \Sigma_b \theta)} = \begin{cases} \arg\min_\theta \frac{\theta^\top \Sigma_w \theta}{\theta^\top \Sigma_b \theta} & \alpha(1 - 6\alpha) \leq 0 \\ \arg\min_\theta \frac{\theta^\top \Sigma_b \theta}{\theta^\top \Sigma_w \theta} & else \end{cases} \tag{10}$$

Note that in the regime where $\alpha(1 - 6\alpha) \leq 0$, $\frac{1-\sqrt{\frac{1}{3}}}{2} \leq p \leq \frac{1+\sqrt{\frac{1}{3}}}{2}$. $\qquad\square$

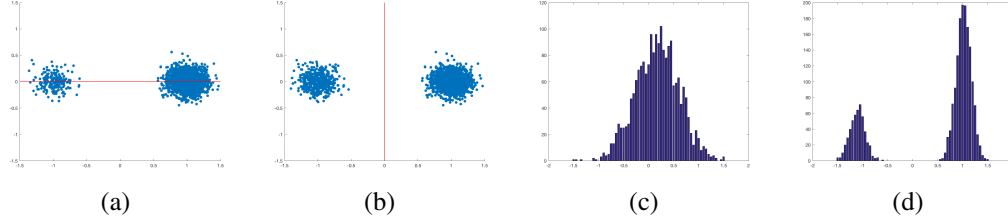

|  (a)  |  (b)  |  (c)  |  (d)  |

Figure 2: A single linear unit trained with VCL and no other loss on 2D inputs. (a) a GMM with $p = 0.1$. (b) a GMM wioth $p = 0.25$. (c) the activations of the learned neuron on the input in (a). (d) similarly for (b). The red lines in (a) and (c) represent the learned projection. For case (a), since for $p = 0.1 < 0.2113$, the projection is such that the two clusters unite. In cases (b), the projection provides a perfect discrimination between the clusters.

This can be interpreted as follows: if both clusters have a relatively high prior probabilities, then the weights of the unit will encourage a separation in the LDA sense. If one cluster has a small prior probability, then the weights will encourage to merge the clusters together by increasing $\theta^\top \Sigma_w \theta$, and decreasing $\theta^\top \Sigma_b \theta$. See Fig. 2. This might be beneficial for preventing overfitting on outliers in the training set, since artifacts that are specific to a small number of training examples have a small prior probability, and will be discouraged from propagating forward.

## 2.2 A Loss for Stochastic Gradient Descent

We now define an alternative regularization based on two mini-batches, and prove a minimum upper bound. Given two sets of iid samples $s_1 = \{\rho_1...\rho_n\}, s_2 = \{\rho'_1...\rho'_n\}$, we define loss variant:

$$L_{s_1,s_2}(p) = \left(1 - \frac{\sigma^2_{s_1}}{\sigma^2_{s_2}}\right)^2 \tag{11}$$

The following theorem shows an upper bound on the deviation of the ratio $\frac{\sigma^2_{s_1}}{\sigma^2_{s_2}}$ from 1.

**Theorem 3.** *It holds that for every* $1 > \epsilon > 0$*:*

$$Pr\left(\frac{4\epsilon^2}{(1+\epsilon)^2} \le (1 - \frac{\sigma^2_{s_1}}{\sigma^2_{s_2}})^2 \le \frac{4\epsilon^2}{(1-\epsilon)^2}\right) \ge \left(1 - \frac{1}{\epsilon^2}(\frac{\kappa(\rho)}{n} - \frac{(n-3)}{n(n-1)})\right)^2 \tag{12}$$

*Proof.* From Chebyshev's inequality, it holds that for any set of iid samples $s = \{x_1...x_n\}$:

$$Pr\left(|1 - \frac{\sigma^2_s}{\mathbb{E}[\sigma^2_s]}| > \epsilon\right) \le \frac{var(\sigma^2_s)}{\epsilon^2(\mathbb{E}[\sigma^2_s])^2} \tag{13}$$

and so with probability of at least $1 - \frac{var(\sigma^2_s)}{\epsilon^2(\mathbb{E}[\sigma^2_s])^2}$ it holds that $1 - \epsilon \le \frac{\sigma^2_s}{\mathbb{E}[\sigma^2_s]} \le 1 + \epsilon$. for two iid sets $s_1, s_2$ with $var(\sigma^2_{s_1}) = var(\sigma^2_{s_2})$ and $\mathbb{E}[\sigma^2_{s_1}] = \mathbb{E}[\sigma^2_{s_2}] = \sigma^2$ we have that:

$$Pr\left(1 - \epsilon \le \frac{\sigma^2_{s_1}}{\mathbb{E}[\sigma^2_{s_1}]}, \frac{\sigma^2_{s_2}}{\mathbb{E}[\sigma^2_{s_2}]} \le 1 + \epsilon\right) \ge \left(1 - \frac{var(\sigma^2_{s_1})}{\epsilon^2\sigma^4}\right)^2 \tag{14}$$

The bound for the ratio follows naturally:

$$Pr\left(\frac{1-\epsilon}{1+\epsilon} \le \frac{\sigma^2_{s_1}}{\sigma^2_{s_2}} \le \frac{1+\epsilon}{1-\epsilon}\right) \ge \left(1 - \frac{var(\sigma^2_{s_1})}{\epsilon^2\sigma^4}\right)^2 \tag{15}$$

and:

$$Pr\left(\frac{4\epsilon^2}{(1+\epsilon)^2} \le (1 - \frac{\sigma^2_{s_1}}{\sigma^2_{s_2}})^2 \le \frac{4\epsilon^2}{(1-\epsilon)^2}\right) \ge \left(1 - \frac{var(\sigma^2_{s_1})}{\epsilon^2\sigma^4}\right)^2 \tag{16}$$

Replacing $var(\sigma^2_{s_1})$ with Eq. 1 completes the proof. $\square$

Note that the RHS of Eq. 12 is maximized when $\kappa(\rho)$ is minimized, similarly to Thm. 1.

In practice, the regularization used during training must be robust to instances where $\sigma_{s_2}^2 \approx 0$, and so the variance constancy loss (VCL) we advocate for is

$$L_{s_1,s_2}^{\beta}(p) = (1 - \frac{\sigma_{s_1}^2}{\sigma_{s_2}^2 + \beta})^2 \tag{17}$$

for some $\beta > 0$. This modification has a two-fold effect. It both stabalizes the loss by preventing exploding gradients, and it encourages the variance for each neuron output to grow. The latter is due to the fact that, by multiplying the activations by a constant scale larger than one, $\beta$ becomes more insignificant. In other words, for $\beta = 0$ the distance between the peaks of the distribution is non-consequential. As $\beta$ grows, there is a stronger driving force that separates the two modes. In our experiments, in order to avoid searching for global optimal values of $\beta$, and since the optimal $\beta$ can vary between layers and neurons, we optimize for this value per-neuron. This is reminiscent to the per-neuron fitting of the additive and multiplicative values in batchnorm.

Note that optimizing $m_4$ directly is not advisable, since estimating higher moments from small batches is prone to large estimation errors.

## 2.3   Batchnorm as a Minimizer of Kurtosis

The use of batchnorm during training of neural networks has been shown to improve test performance, as well as speed up training time. In batchnorm, sample statistics of each mini-batch are calculated, and used for normalization of the activations (either before or after the application of non-linearity). Specifically, each activation is normalized to have zero mean and unit variance. This scheme introduces additional randomness in the network, since the output of a unit depends on the particular mini-batch statistics, as well as the particular input sample. Since the sample mean is a much more reliable statistic than the sample variance, most of the randomness is caused by the variance of the sample variance.

Consider a single unit $\rho_x$ that undergoes batch-norm during training. The output of that unit given input $x$ and batch $s$ is given by $\frac{\rho(x)-\mu_s}{\sigma_s}$. We expect the batch statistics $\sigma_s, \mu_s$ to be reliable approximations of the actual statistics, otherwise performance would vary wildly between test and train splits, as well as between mini-batches during training. We therefore expect for each sample $x$:

$$\left| \frac{\rho(x)-\mu_s}{\sigma_s} - \frac{\rho(x)-\mu}{\sigma} \right| = \left| \frac{\rho(x)-\mu}{\sigma} \right| \left| \frac{\sigma}{\sigma_s} \frac{\rho(x)-\mu_s}{\rho(x)-\mu} - 1 \right| << 1 \tag{18}$$

We note that $\mu_s$ is a more reliable statistic than $\sigma_s$, and so $\frac{\rho(x)-\mu_s}{\rho(x)-\mu} \approx 1$. Since this applies to all inputs $x$, we have:

$$\left| \frac{\sigma}{\sigma_s} - 1 \right| << 1, \frac{\sigma}{\sigma_s} \approx 1 \tag{19}$$

From Chebyshev's inequality, it holds for $1 > \epsilon > 0$:

$$Pr\left( \frac{1}{\sqrt{1+\epsilon}} \le \frac{\sigma}{\sigma_s} \le \frac{1}{\sqrt{1-\epsilon}} \right) \ge 1 - \frac{var(\sigma_s^2)}{\epsilon^2 (\mathbb{E}[\sigma_s^2])^2} = 1 - \frac{1}{\epsilon^2} \left( \frac{\kappa(\rho)}{n} - \frac{(n-3)}{n(n-1)} \right) \tag{20}$$

Therefore, under mild assumptions, a low value for the Kurtosis leads to a stable application of batchnorm. Note that in batchnorm, Eq. 18 is not forced, and so kurtosis is not explicitly minimized.

## 2.4   The Loss in Action

According to Thm. 3, when the sample size $n$ is large, the bound on the probability in the RHS of Eq.12 is high regardless of $\kappa$. Therefore, the ratio of the sample variance and the true variance is close regardless of the shape of the distribution. This favors small $n$ for the VCL method. Empirically, we notice that VCL tends to work better as $n$ is lower, where the best results for CNN models are achieved when setting $n = 2$.

We opt for the simplest way to sample minibatches of size $n$ for the loss, without changing the mini-batches that are used for the SGD procedure. Assume that the size of the SGD minibatches

is $N$. Typically $n < 2N$, and we take out of the $N$ samples of the SGD minibatch the first two consecutive subsets of size $n$. The variance constancy loss (VCL) is computed based on these two arbitrary subsets. In all of our experiments $\beta$ is set to an initial value of $1.0$, and then updated for each unit through backpropagation. In our experiments, the VCL terms are averaged in each layer, and then summed up across layers. A weight $\gamma$ is applied to this loss.

When $n$ is very small, training becomes unstable due to increasing random variations in sample statistics. This instability is minimized by VCL, which increases its overall influence. In order to support such small $n$, training is stabilized by performing gradient clipping. Specifically, the L2 norm of the gradient of each layer is clipped, with a clipping value of 1.

## 3 Experiments

Comparing different activation functions or different normalization schemes and their combinations, is a notorious task: every choice benefits the most from a different set of hyperparameters, leading to large search space and high computational demands and, often, reproducibility issues. The authors of [11], for example, provided an exemplary set of experiments to demonstrate that their SeLU activation function outperforms other activation functions. For the UCI datasets, the authors provide detail experimental protocols, some code, and all the train/test splits. Despite all these, we were not able to completely replicate their UCI experiments for various reasons. First, our resources allowed us to test less architectures by the deadline. Second, we were uncertain regarding, for example, the amount and location of dropout used. In another example, we were able to replicate the CIFAR experimental result for the ELU activation function [2]. However, unlike the published results, in our experiments, batchnorm improves the accuracy. This highlights the challenges of comparative experiments, but is in no way a criticism on the previous work. Indeed, both ELU and SeLU have provided a great deal of performance gain in a wide variety of follow-up work.

We demonstrate the effectiveness of VCL regularization on several benchmark datasets, comparing with competitive baselines. We conduct two sets of experiments. In the first set of experiments, we test CNNs on the CIFAR-10, CIFAR-100 and tiny Imagenet datasets. In the second, we evaluate fully connected networks on all of the UCI datasets with more than 1000 samples. To support reproducibility, the entire code of all of our experiments is to be promptly released.

**CIFAR** The two CIFAR datasets (Krizhevsky Hinton, 2009) consist of colored natural images sized at 32×32 pixels. CIFAR-10 (C10) and CIFAR-100 (C100) images are drawn from 10 and 100 classes, respectively. For each dataset, there are 50,000 training images and 10,000 images reserved for testing. We use a standard data augmentation scheme (Lin et al., 2013; Romero et al., 2014; Lee et al., 2015; Springenberg et al., 2014; Srivastava et al., 2015; Huang et al., 2016b; Larsson et al., 2016), in which the images are zero-padded with 4 pixels on each side, randomly cropped to produce 32×32 images, and horizontally mirrored with probability 0.5.

For the CIFAR datasets, we employ the 11-layer architecture that was used by [2] to compare activation functions. The 18-layer architecture was trained with a dedicated dropout scheduling that makes it more specific to a certain choice of activation function, and is slower to train. We do not employ ZCA whitening on the data since it seems to decrease the overall accuracy for ReLU and Learky ReLU. For all experiments, 500 epochs are used and a batch size $N$ of 250. We employ a learning rate of 0.05, which was reduced at epoch 180 to 0.02, and further reduced by a factor of 10 every 100 epochs. A momentum of 0.9 was used and the L2 regularization term was weighed by 0.0001. The hyperparameters of VCL are fixed: the weight of the VCL regularization is set to $\gamma = 0.01$.

The results are presented in Tab. 2, with running time per training iterations comparisons presented in Tab. 1. We compare ReLU to Leaky ReLU with a constant of 0.2 and to ELU, with different normalization techniques. Experiments with VCL are performed with $n = 2, 3, 5, 7, 9$. Our result for CIFAR-100 of the ELU activation matches the reported result in [2] (CIFAR-10 result is not provided for this architecture). As can be seen, batchnorm contributes to ReLU and ELU but not to Leaky ReLU. The best results are obtained with a combination of ELU and our VCL method for both datasets. The only experiment in which VCL does not contribute more than batchnorm is the ReLU experiment on CIFAR-100. The largest contribution of VCL is to ELU.

Table 1: Time in Seconds per 100 iterations (CIFAR-100).

| Method | Intel i7 CPU | Volta GPU |
|---|---|---|
| Without normalization | 367.1 | 29.2 |
| Batchnorm | 702.3 | 31.6 |
| VCL | 400.1 | 30.3 |

Table 2: Test error w/o normalization, with bathnorm (bn), layer normalization (ln), group normalization (gn) or vcl.

| | CIFAR-10 | CIFAR-100 | | CIFAR-10 | CIFAR-100 |
|---|---|---|---|---|---|
| ReLU | 0.0836 | 0.328 | LReLU+vcl ($n = 9$) | 0.0660 | 0.267 |
| ReLU+bn | **0.0778** | **0.291** | LReLU+vcl ($n = 7$) | 0.0665 | 0.264 |
| ReLU+ln | 0.0792 | 0.307 | LReLU+vcl ($n = 5$) | 0.0648 | 0.264 |
| ReLU+gn | 0.0871 | 0.319 | LReLU+vcl ($n = 3$) | 0.0657 | **0.262** |
| ReLU+vcl ($n = 9$) | 0.0780 | 0.308 | LReLU+vcl ($n = 2$) | **0.0645** | 0.263 |
| ReLU+vcl ($n = 7$) | 0.0810 | 0.305 | | | |
| ReLU+vcl ($n = 5$) | 0.0785 | 0.304 | ELU | 0.0698 | 0.287 |
| ReLU+vcl ($n = 3$) | 0.0790 | 0.306 | ELU+bn | 0.0663 | 0.269 |
| ReLU+vcl ($n = 2$) | 0.0780 | 0.303 | ELU+ln | 0.0675 | 0.267 |
| | | | ELU+gn | 0.0671 | 0.282 |
| LReLU | 0.0670 | 0.268 | ELU+vcl ($n = 9$) | 0.0670 | 0.276 |
| LReLU+bn | 0.0708 | 0.272 | ELU+vcl ($n = 7$) | 0.0633 | 0.271 |
| LReLU+ln | 0.0700 | 0.270 | ELU+vcl ($n = 5$) | **0.0615** | 0.258 |
| LReLU+gn | 0.0707 | 0.283 | ELU+vcl ($n = 3$) | 0.0622 | 0.261 |
| (Continued to the right) | | | ELU+vcl ($n = 2$) | **0.0615** | **0.256** |

**Tiny Imagenet** The Tiny ImageNet dataset consists of a subset of ImageNet [16], with 200 different classes, each of which has 500 training images and 50 validation images, downscaled to $64 \times 64$. For augmentation, the images are zero padded with 8 pixels on each side, and randomly cropped to produce $64 \times 64$ images, and then horizontally mirrored with probability 0.5.

For this set, we employ a similar architecture used for the CIFAR experiments, with twice as many convolutional kernels per layer. In order to account for the higher resolution images, we apply average pooling at the end of the 5'th convolutional block. We also use the same hyper parameters as in the CIFAR experiments, namely $\gamma = 0.01$, and $n = 5$. A learning rate of 0.05 is employed, which is reduced to 0.02 after 50 epochs, and further reduced by 10 at 100 and 180 epochs. We report the validation accuracy after 250 epochs. The results are reported in Tab. 3. Results for Resnet-110, WRN-32, DenseNet-40 are as reported in [6].

**UCI** We also apply VCL to the 44 UCI datasets with more than 1000 samples. The train/test splits were provided by the authors of [11]. In each experiment, we three fixed architectures with 256 hidden neurons per layer and depth of either 4, 8, or 16. For ReLU and ELU the last layer had a dropout rate of 0.5. For SeLU, we employ the prescribed $\alpha-$dropout rate of 0.05 for all hidden layers. A learning rate of 0.01 was used for the first 200 epochs and then a learning rate of $10^{-3}$ was used. All runs were terminated after 500 epochs. Following [11], an averaging operator with a mask size of 10 was applied to the validation error, and the epoch and architecture with the best smoothed

| | Validation error | | Validation error |
|---|---|---|---|
| Deep ELU network | 0.392 | ResNet-110 | 0.465 |
| Deep ELU network + bn | 0.402 | Wide-ResNet-32 | **0.365** |
| Deep ELU network + vcl n=2 | **0.373** | DenseNet-40 | 0.390 |

Table 3: Validation error on tiny imagenet. We ran the three Deep ELU experiments. The baseline results are from [6].

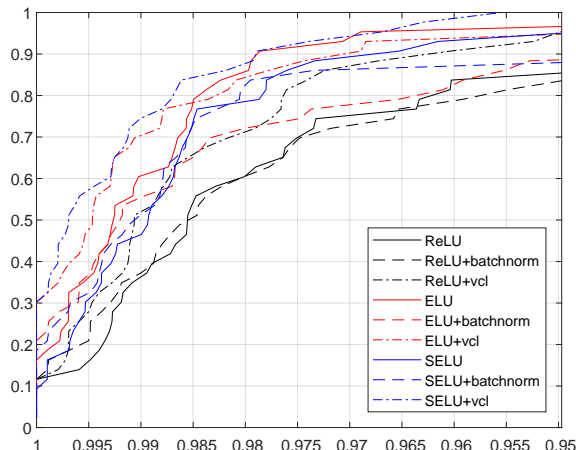

Figure 3: An accuracy based Dolan-More profile for the UCI experiments of Tab. 5. There are 9 plots, one for each combination of activation and normalization. The x-axis is the threshold ($\tau$). Since for accuracy scores, higher is better, whereas typical Dolan-More plots show cost (such as run-time), the axis is ordered in reverse. The y-axis is, for a given combination out of the 9, the ratio of datasets in which the obtained accuracy is above $\tau$ times the maximal accuracy over all 9 options.

Table 4: Number of "wins" for each normalization method, per activation function.

|                    | ReLU | ELU | SELU |
|--------------------|------|-----|------|
| No normalization   | 9    | 14  | 11   |
| Batchnorm          | 15   | 16  | 15   |
| VCL                | **27** | **23** | **28** |

validation error was selected. Batches were of size $N = 20$, $\gamma = 0.01$, and, for these experiments, $n = 10$.

The results are shown in Fig. 3 and fully reported in the appendix (Tab. 5). As expected, no method wins across all experiments. However, the results show that the method that wins the most (out of the 9 options) is either the combination of SeLU and VCL or that of ELU and VCL. A breakdown per each activation unit separately is presented in Tab. 4. A win is counted if the method reaches the minimal value among the three normalization options and if performance is not constant. For all three activation functions, VCL provides more wins than batchnorm, and batchnorm outperforms the no normalization option. The gap between VCL and batchnorm is larger for SELU and the lowest for ReLU, which is also consistent with the results in Tab. 2.

## 4 Related Work

The seminal batchnorm method [8] has enabled a markable increased in performance for a great number of machine learning tasks, ranging from computer vision [5] to playing board games [20]. In practice, the method is said to suffer from a few limitations [17, 7, 24]. One of these limitations is the reliance on the batch statistics during the forward step, including at test time, which is performed one sample at a time. The training statistics are therefore used as surrogates at test time, which is detrimental as there is a shift between the training and the test distributions [14]. Our method, as a loss-based method, does not employ batch statistics at test time.

Another limitation of batchnorm is the reliance on batch statistics, which are unreliable for small batches. This leads to the need to employ larger batches, which tend to result in worse generalization [24]. This disadvantage turns into an advantage in our method, since this instability is what our method aims to reduce. Indeed, we perform our experiments with only a few samples for the VCL loss.

Other normalization techniques, which do not rely on batch statistics include classical methods, such as local response normalization [13, 9, 12], layer normalization [1], instance normalization [22], weight normalization [18], and the very recent group normalization [24].

Since our regularization term encourages bimodal activation distributions, it is somewhat related to the study of networks with binary activation functions [3]. However, our goal is to increase the classification accuracy and not to achieve the efficiency benefits of binary activations.

Considering one of the modes as a baseline activation, our work can be viewed as related to sparsity regularization methods, including L1 regularization [21] and its local or selective application [19, 25] and structural sparsification methods [23] that also modify the architecture by pruning some of the connections. Such methods lead to more efficient networks as well as to an improvement in accuracy.

Our method is also related to variational methods such as the variational autoencoder [10], which employs a regularization term that enforces a certain distribution on some of the activations. The target distribution is often taken to be Gaussian in contract to our loss term that encourages multiple modes. In this sense, our work is more related to discrete variational autoencoders [15]. In contrast to such work, our method employs the regularization term everywhere, the multi-modal structure is soft, and the number of modes is not enforced (Thm. 2, and the fact that multi-peak distributions with more than 2 peaks also have low Kurtosis).

## 5 Conclusions

The batchnorm method plays a pivotal role in many of the recent successes of deep learning. With the growing dependency on this method, some researchers have voiced concerns about the required batch sizes. VCL employs small subsets of the mini-batch and seems to perform as well or better than batchnorm on the standard benchmarks tested. It therefore holds the promise of improving conditioning without imposing constraints on the optimization process. Since VCL is a regularization term and not a normalization mechanism, and since the statistics of sample moments is well understood, the new method could be compatible with a wider variety of optimization methods in comparison to bachnorm. Compared to other loss terms, VCL shapes the activation distribution in one of several phases, according to the input statistics.

As future work, we would like to address some limitations that were observed during the experiments. The first is the observation that while VCL shows good results with the ReLU activations on the UCI experiences, in image experiments the combination of the two underperforms when compared to ReLU with batchnorm. The second limitation is that so far we were not able to replace batchnorm with VCL for ResNets.

## Acknowledgements

This project has received funding from the European Research Council (ERC) under the European Union's Horizon 2020 research and innovation programme (grant ERC CoG 725974).

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

# A  More results

Table 5: The results of the UCI experiments

| | ReLU | | | ELU | | | SeLU | | |
|---|---|---|---|---|---|---|---|---|---|
| | | bn | vcl | | bn | vcl | | bn | vcl |
| abalone | 0.334 | 0.342 | 0.331 | **0.325** | 0.342 | 0.330 | 0.343 | 0.335 | 0.339 |
| adult | 0.156 | 0.148 | 0.155 | 0.152 | 0.148 | 0.155 | 0.150 | 0.148 | **0.147** |
| bank | 0.112 | 0.108 | 0.109 | 0.103 | 0.106 | **0.099** | 0.112 | 0.110 | 0.107 |
| car | 0.054 | 0.029 | 0.041 | 0.039 | **0.018** | 0.036 | 0.031 | 0.032 | 0.025 |
| cardio.-10clases | 0.221 | 0.238 | 0.224 | 0.214 | 0.223 | 0.204 | 0.219 | 0.211 | **0.202** |
| cardio.-3clases | 0.106 | 0.110 | **0.096** | 0.108 | 0.108 | 0.104 | 0.103 | 0.109 | 0.097 |
| chess-krvk | 0.250 | 0.301 | 0.233 | 0.217 | 0.307 | **0.207** | 0.226 | 0.435 | 0.218 |
| chess-krvkp | 0.027 | 0.015 | 0.023 | 0.020 | **0.009** | 0.016 | 0.010 | 0.010 | 0.010 |
| connect-4 | 0.146 | 0.153 | 0.150 | 0.153 | **0.139** | 0.143 | 0.143 | 0.144 | **0.139** |
| contrac | 0.502 | 0.546 | 0.480 | 0.490 | 0.506 | 0.490 | 0.475 | 0.501 | **0.454** |
| hill-valley | 0.530 | 0.399 | 0.270 | 0.272 | 0.268 | 0.301 | 0.276 | 0.349 | **0.187** |
| image-segmentation | 0.006 | **0** | 0.006 | 0.006 | 0.006 | 0.012 | 0.012 | 0.012 | 0.012 |
| led-display | 0.309 | 0.326 | 0.307 | 0.295 | 0.319 | 0.295 | 0.299 | **0.290** | 0.305 |
| letter | 0.061 | 0.038 | 0.051 | 0.054 | **0.037** | 0.044 | 0.043 | **0.037** | 0.045 |
| magic | 0.138 | 0.135 | 0.133 | 0.138 | 0.131 | 0.130 | 0.130 | **0.125** | 0.126 |
| miniboone | 0.084 | 0.090 | 0.083 | 0.075 | 0.070 | 0.073 | 0.081 | **0.068** | 0.080 |
| molec-biol-splice | 0.214 | 0.223 | 0.205 | 0.18 | 0.192 | 0.189 | 0.172 | **0.163** | 0.194 |
| mushroom | 0 | 0 | 0 | 0 | 0 | 0 | 0 | 0 | 0 |
| nursery | 0.007 | 0.005 | 0.005 | 0.001 | 0.004 | **0** | **0** | 0.006 | **0** |
| oocytes-m.-nucleus-4d | 0.228 | 0.209 | 0.196 | 0.205 | 0.194 | 0.202 | 0.199 | **0.181** | 0.184 |
| oocytes-m.-states-2f | 0.091 | 0.096 | 0.093 | 0.090 | 0.09 | **0.085** | 0.097 | 0.088 | 0.093 |
| optical | 0.039 | **0.025** | 0.033 | 0.034 | 0.026 | 0.032 | 0.040 | 0.030 | 0.032 |
| ozone | **0.028** | 0.033 | 0.031 | 0.031 | 0.036 | 0.029 | 0.029 | 0.047 | 0.031 |
| page-blocks | 0.039 | 0.037 | 0.039 | 0.039 | 0.042 | **0.032** | 0.033 | 0.033 | 0.036 |
| pendigits | 0.043 | 0.041 | 0.037 | 0.044 | 0.040 | 0.038 | 0.041 | **0.035** | 0.037 |
| plant-margin | 0.291 | 0.305 | 0.282 | **0.280** | 0.314 | 0.296 | 0.305 | 0.321 | 0.281 |
| plant-shape | 0.433 | 0.442 | 0.420 | 0.393 | 0.462 | **0.387** | 0.419 | 0.463 | 0.403 |
| plant-texture | 0.297 | 0.281 | 0.297 | 0.273 | 0.279 | 0.282 | 0.278 | 0.283 | **0.268** |
| ringnorm | 0.022 | 0.026 | 0.021 | 0.021 | 0.025 | **0.018** | 0.025 | 0.035 | 0.021 |
| semeion | 0.116 | 0.110 | 0.111 | 0.105 | **0.103** | 0.107 | 0.112 | 0.115 | 0.115 |
| spambase | 0.075 | 0.075 | 0.068 | 0.070 | **0.063** | 0.068 | 0.066 | 0.069 | 0.070 |
| statlog-german-credit | 0.296 | 0.273 | 0.248 | 0.252 | 0.289 | **0.228** | 0.245 | 0.243 | 0.242 |
| statlog-image | 0.045 | 0.041 | 0.047 | 0.051 | **0.038** | 0.04 | 0.041 | 0.044 | 0.040 |
| statlog-landsat | 0.114 | 0.113 | 0.106 | 0.108 | 0.110 | 0.106 | **0.095** | 0.104 | 0.100 |
| statlog-shuttle | 0.001 | 0.001 | 0.001 | 0.001 | 0.001 | 0.001 | 0.001 | 0.004 | 0.001 |
| steel-plates | 0.299 | 0.280 | **0.270** | 0.285 | 0.276 | 0.274 | 0.281 | 0.276 | 0.272 |
| thyroid | 0.026 | 0.024 | 0.021 | 0.020 | 0.021 | **0.017** | 0.021 | 0.024 | 0.019 |
| titanic | **0.208** | **0.208** | **0.208** | **0.208** | **0.208** | **0.208** | 0.214 | 0.215 | **0.208** |
| twonorm | 0.030 | 0.040 | 0.028 | 0.028 | 0.029 | 0.029 | 0.026 | 0.027 | **0.025** |
| wall-following | 0.126 | 0.115 | 0.112 | 0.106 | 0.105 | **0.093** | 0.103 | 0.104 | 0.102 |
| waveform-noise | 0.173 | 0.197 | 0.172 | 0.164 | 0.164 | 0.163 | 0.162 | 0.163 | **0.153** |
| waveform | 0.164 | 0.166 | 0.167 | 0.149 | 0.164 | 0.161 | 0.151 | 0.160 | **0.147** |
| wine-quality-red | 0.413 | 0.414 | 0.404 | **0.397** | 0.424 | 0.431 | 0.432 | 0.413 | 0.417 |
| wine-quality-white | **0.461** | 0.483 | 0.468 | **0.461** | 0.482 | 0.478 | 0.469 | 0.491 | 0.485 |
| Number of wins out of 9 options | 3 | 3 | 3 | 5 | 8 | **11** | 2 | 7 | **11** |

