[Reviews · NeurIPS 2018]

Reviewer 1



Overview: The paper revisits the notion of variance regularization, where instead of a renormalization layer, a loss term is added to prevent fluctuations in batch statistics across minibatches. The authors present numerical results, connect the idea to BatchNormalization, and show theoretical properties of their approach in restricted cases. Strengths: The problem being studied is of great importance to the community. Renormalization strategies such as BatchNorm or LayerNom have shown to greatly speed up training of deep networks but are poorly understood theoretically. Further, in many cases such as low mini-batch, GAN training, reinforcement learning, these strategies might do more harm than good and knowing that a-priori is difficult if not impossible. The authors propose a loss-based regularization instead of architectural changes and share interesting theoretical insights about this approach. The numerical results presented are comprehensive and I especially liked the UCI experiments. As means to aggregate information in such experiments, I suggest that the authors look at Dolan-More profiles which are ubiquitous in the nonlinear optimization community. Weaknesses: - What is the time comparison of VCL relative to BatchNorm and having no normalization? - The argument tying modes, BatchNorm, and VCL could be better explained. It seems that the observations about modes and normalization outcome is new but the authors don't describe it sufficiently. - I recommend that the authors format their mathematical equations better. For instance, Equations (4), (14), (18), and others, would be easier to parse if the bracketing and indexing were fixed. - Line 177 typo: "batchsized" - It would aid a reader if the authors summarized the loss and how it is computed at the end of Section 2.2. - How sensitive is the framework to the choice of n? - How does \beta vary over time? Could the authors include a graph for this in the Appendix? Questions: - Will the authors be open-sourcing the code for the experiments? - Have you experimented with a constant \beta? - Have you experimented with having _both_ BN and VCL? Post-rebuttal I will stick to my rating. This is good work and I thank the authors for clarifying my questions in the rebuttal.

Reviewer 2



# Post rebuttal I believe it is a good submission (7), with some of the issues raised alraedy addressed by the authors rebuttal. # Original review This paper describes a new local regulariser for neural networks training to, in principle, reduce the variance of the variance of the activations. Pros: - an easy to follow line of thought - reasonable mathematical justification of the proposed method - empirical evaluation showing results comparable with batch norm, but without all the problems BN introduces (tracking statistics, heavily affected by batch size, unclear transfer to test time etc.) - A new view of BN as a Kurtosis minimiser (while it could be obvious for some mathematicians, it is definitely a nicely described relation) - Proposed method introduces interesting multi-modality effect to neuron activations Cons: - Evaluation is minimalistic, while batch norm is indeed extremely popular, it is not the state-of-the-art normaliser, methods like layer normalisation etc. has been shown to outperform it too, consequently it is not clear if proposed method would compare favourably or not. In reviewer's opinion, it is worth considering this paper even if the answer would be negative, as it is an alternative approach, focus on loss, rather than forward pass modification, and as such is much more generic, and "clean" in a sense. - Change from (6) to (11) is not well justified in the text Minor comments: - Please use \left and \right operators in equations so that brackets are of correct size (for example in (12) or (13)) - Figures require: a) bigger labels / ticks b) more precise captions, in Figure 1 what do columns represent? - please change "<<" to "\ll" which is latex command for this symbol - Figure 1 would benefit from having third panel with activation of a network without any of the two techniques

Reviewer 3



This paper introduces Variance Constancy Loss (VCL), a novel normalization technique for DNNs. VCL incorporates regularizer loss which penalizes variance of activation’s variance of a DNN layer. VCL allows achieving results which are better or comparable to those of Batch Normalization. These results look very interesting given the fact that VCL adds only a regularizing term and does not modify model workflow as BN does. Another benefit of VCL is that compared to BN it does not require large batches for consistency. The experimental evaluation covers variety of UCI tasks for fully-connected networks and CIFAR-10/CIFAR-100 data for deep CNN architecture. The VCL is shown to perform better or comparable to BN in terms of final test error. However, it is not clear whether VCL speeds up the training which is a desirable feature of a normalization technique. The paper is clearly written and easily readable. Although, it seems that there is a typo on line 146 in the inequality “n < 2N”. Instead, there should be “N < 2n” as in the following text it is said that VCL extracts two consecutive subsets of size n from a mini-batch of size N. Overall, the paper is good. It proposes a novel normalization technique with useful properties which are theoretically grounded and are shown to be effective in practice. Edits after Author Response: I keep my rating. The authors have addressed my question in the rebuttal.